# Muscle Tone Assessment by Machine Learning Using Surface Electromyography

**DOI:** 10.3390/s24196362

**Published:** 2024-09-30

**Authors:** Andressa Rastrelo Rezende, Camille Marques Alves, Isabela Alves Marques, Luciane Aparecida Pascucci Sande de Souza, Eduardo Lázaro Martins Naves

**Affiliations:** 1Assistive Technology Laboratory, Faculty of Electrical Engineering, Federal University of Uberlandia, Uberlandia 38400-902, Brazil; camillealves@ufu.br (C.M.A.); isabelamarquesvj@gmail.com (I.A.M.);; 2Department of Applied Physical Therapy, Federal University of Triangulo Mineiro, Uberaba 38065-430, Brazil; luciane.souza@uftm.edu.br

**Keywords:** neurological disorders, muscle tone, evaluation, surface electromyography, machine learning, classification

## Abstract

Muscle tone is defined as the resistance to passive stretch, but this definition is often criticized for its ambiguity since some suggest it is related to a state of preparation for movement. Muscle tone is primarily regulated by the central nervous system, and individuals with neurological disorders may lose the ability to control normal tone and can exhibit abnormalities. Currently, these abnormalities are mostly evaluated using subjective scales, highlighting a lack of objective assessment methods in the literature. This study aimed to use surface electromyography (sEMG) and machine learning (ML) for the objective classification and characterization of the full spectrum of muscle tone in the upper limb. Data were collected from thirty-nine individuals, including spastic, healthy, hypotonic and rigid subjects. All of the classifiers applied achieved high accuracy, with the best reaching 96.12%, in differentiating muscle tone. These results underscore the potential of the proposed methodology as a more reliable and quantitative method for evaluating muscle tone abnormalities, aiming to address the limitations of traditional subjective assessments. Additionally, the main features impacting the classifiers’ performance were identified, which can be utilized in future research and in the development of devices that can be used in clinical practice.

## 1. Introduction

Muscle tone has been defined by Sanger et al. [1] as “the resistance to passive stretch while a patient attempts to maintain a relaxed state of muscle activity”. This concept is widely used and accepted within this field. However, Shortland [2] has raised concerns, arguing that the definition lacks clarity and leaves room for ambiguity. This ambiguity can lead therapists to develop their own interpretations of what constitutes resistance to movement, resulting in interrater variability and increasing the subjectivity of assessments. Bernstein et al. [3] offer a different perspective, suggesting that muscle tone is related to a state of preparation for movement and cannot be accurately assessed when a person is asked to relax and not move the limb.

Muscle tone is primarily regulated by the central nervous system (CNS) but is also influenced by the peripheral nervous system (PNS). When individuals have neurological disorders, this regulation can be disrupted, leading to an inability to control muscle tone normally, which can result in abnormalities [4]. These abnormalities can be related to decreases in muscle tone (hypotonia) or unwanted increases in tone (hypertonia). The two main types of hypertonia are spasticity (elastic hypertonia) and rigidity (plastic hypertonia). The relationship between the physiological states of muscle tone is crucial for understanding motor dysfunctions. Normal tone represents the ideal state of slight resistance, while hypotonia reflects a decrease in this tone, resulting in flaccid muscles and difficulties in movement [5]. In contrast, spasticity and rigidity are forms of hypertonia, where spasticity is characterized by an abnormal increase in resistance [6], especially in response to rapid movements, and rigidity presents constant resistance [7].

Abnormalities in muscle tone can be highly disabling, resulting in pain, joint contractures and, consequently, increased healthcare costs [8]. Since the upper limbs play an essential role in daily life—contributing to postural balance, proper gait and the performance of everyday tasks, such as eating and dressing—any alteration in muscle tone in these limbs affects all of these functions. This can lead to increased dependence on the individual, which may subsequently affect their overall quality of life.

The assessment of muscle tone is a crucial factor in clinical diagnosis and the monitoring of treatment outcomes. Accurately measuring muscle tone allows clinicians to evaluate the effectiveness of treatments and make informed decisions about patient care. Muscle tone is usually assessed using passive resistance to movement, using ordinal scales such as the Modified Ashworth Scale (MAS) [9], Tardieu Scale [10] and Unified Parkinson’s Disease Rating Scale (UPDRS) [11]. These ordinal scales are a matter of discussion because they are assessments that depend on the evaluator’s experience and cannot present an objective evaluation of muscle tone [12].

Several studies have developed methods to objectively assess different types of muscle tone. These methods typically utilize kinematic components [13], torque [14], surface electromyography (sEMG) [15,16], tensiomyography [17], myotonometry [18], ultrasound elastography [19] and inertial sensors [20]. Most of these approaches focus on the evaluation of spasticity [14] and rigidity [21]. In contrast, the assessment of hypotonia is less explored compared to other types [22,23]. Another gap in the literature is the lack of studies focusing on identifying and characterizing different types of abnormalities using a single method. The existing research often targets only one type of muscle tone alteration, which could greatly benefit therapists in clinical practice.

Some studies also utilize machine learning (ML) algorithms for the evaluation and diagnosis of alterations in muscle tone. ML is a branch of Artificial Intelligence (AI) that combines ideas from various fields, such as neuroscience, biology, statistics and mathematics, to separate and classify complex patterns in the collected data [24]. The work of [20] used sEMG, inertial sensors and ML to assess spastic subjects based on their level of impairment and achieved high accuracy as a result.

In this context, the primary objective of this study was to develop and evaluate a methodology for the objective classification and characterization of the full spectrum of muscle tone in the upper limb, including abnormalities. This spectrum ranges from hypertonicity (including spasticity and rigidity) to hypotonicity and normotonicity, using sEMG signals from the biceps and triceps brachii muscles. This study sought to overcome the limitations of subjective ordinal scales by employing machine learning (ML) classifiers, such as the K-Nearest Neighbor (KNN), Support Vector Machine (SVM), Random Forest (RF) and Gradient Boosting Machine (GBM). The goal was to achieve high classification accuracy and reliability, ultimately providing an objective and quantitative tool for the clinical diagnosis and monitoring of muscle tone in individuals with neurological disorders.

## 2. Materials and Methods

### 2.1. Participants

For this study, 39 subjects were recruited, divided into four different groups based on the muscle tone alterations they exhibited: spastic tone (10), hypotonic tone (9), normal tone (10) and rigid tone (10). To be included in this study, individuals needed to meet certain eligibility criteria, some of which were general and others specific to each group. The general eligibility criteria were as follows:Age between 18 and 80 years old;Not to present with other associated musculoskeletal or neurodegenerative diseases;Be clinically stable;Agree and sign the Informed Consent Form.

To compose the spastic group, the participants needed to exhibit spasticity in the biceps and triceps brachii muscles, confirmed by a therapist. For the rigid group, the participants needed to be clinically diagnosed with PD by a neurologist, and exhibit rigidity as one of the cardinal characteristics, evaluated by a therapist. And for the hypotonic group, the participants needed to be diagnosed with hypotonia by a therapist, resulting from neurological disorders, involving CNS and/or PNS impairment. The healthy group was selected with ages and genders compatible with the other groups to maintain the uniformity of the sample. The therapist who conducted the initial assessment and checked the eligibility criteria was the same one who carried out all the data collection as well. On the other hand, the applied exclusion criteria were as follows:Inability to perform the proposed movements;Not understand the instructions provided;

### 2.2. Data Acquisition

To acquire biomedical data, a two-channel surface electromyograph was used, with one electrode positioned on the biceps brachii and the other on the triceps brachii, following the Surface ElectroMyoGraphy for the Non-Invasive Assessment of Muscles (SENIAM) protocol [25]. Additionally, an electrogoniometer was employed to measure the joint angles during movements, enabling precise signal windowing. These devices were custom-developed by our team to ensure precise control over all the production stages, including defining the desired gain range, which reduced the final cost of the equipment and made it more suitable for future clinical applications. AgCl electrodes were used for data collection, positioned parallel to the muscle fibers with a distance of 2 cm between them.

The developed sEMG equipment offers variable gains of 760, 1000 and 1200 times and operates at a sampling frequency of 2000 Hz. The device has already been tested in other applications [26] and has shown good results. The electrogoniometer was developed using Polymer Optical Fiber (POF) due to its advantages over similar devices, including being compact, lightweight, flexible, low cost and immune to electromagnetic interference. This sensor has also been validated in [27] and demonstrates good repeatability.

### 2.3. Experimental Protocol

Each step of the protocol was carefully devised and developed collaboratively with the team and then replicated during the data collection phase for all the participants. This approach aimed to ensure the reliability of the dataset and mitigate potential biases. The initial phase of the data collection protocol involved introducing the research to the participants, elucidating its objectives and explaining the procedural steps. All the collections were performed on only one side of each patient, and the chosen side was, in all cases, the most affected side, i.e., the side with the greatest alteration in muscle tone. For the healthy participants, the data were collected from the dominant side. After the initial stage, the preparation for acquiring biomedical signals began. The participant was comfortably seated with their arm supported on a brace to stabilize movements and maintain the correct joint angles. The support was used to isolate the movement and focus solely on the muscles of interest. Figure 1 shows the position in which the participant was placed to perform the movements.

Next, the skin was cleaned with alcohol and cotton at the sites where the disposable electrodes would be applied. The electrogoniometer was then positioned on the subject’s arm, centered with the elbow joint. After this, the resting signal was collected for 10 s. Then, data collection for the Maximum Voluntary Contraction (MVC) began, involving three repetitions for each muscle, each lasting 5 s with a 30 s rest period between contractions. In the final stage, the participants performed active movements, executing full elbow extensions and flexions at a comfortable pace. Each individual completed 15 repetitions of this exercise.

### 2.4. Data Processing

Figure 2 presents the diagram outlining the steps taken for sEMG signal processing. The processing was conducted using R-Studio with the R language 4.4.1 [28].

#### 2.4.1. Preprocessing

The objective of preprocessing is to remove noise such as motion artifacts, electrical network interferences and variations in the electrode–skin interface impedance to isolate information related solely to muscle activity. To achieve this, filtering was performed using a Butterworth band-pass filter with a range of 20 Hz to 500 Hz and a notch filter at 60 Hz. The Empirical Mode Decomposition (EMD) method was also applied to effectively extract the signal’s features while removing unwanted interferences. Following this, the signal was normalized using the MVC.

#### 2.4.2. Windowing

After preprocessing, the sEMG signal was segmented into windows. During the data collection, the subjects performed 15 repetitions, starting with the arm fully flexed. The participant then fully extended their arm, and the stretch ended when the participant returned the arm to the fully flexed position. Joint angles were used to accurately mark the start and end of each stretch, consistently marking the point of total flexion for each participant.

#### 2.4.3. Feature Extraction

In this study, eleven different features were used, with eight in the time domain and three in the frequency domain. This approach was taken to ensure a comprehensive view of the captured signal, as features in the time domain provide information about the magnitude of and variability in the signal, while those in the frequency domain offer details of the spectral composition and energy distribution. The features selected in the time domain were the Mean Absolute Value (MAV), Modified Mean Absolute Value (MMAV), Standard Deviation (STD), Waveform Length (WL), Zero Crossing (ZC), Slope Sign Change (SSC), Root Mean Square (RMS) and Log Detector (LD). In the frequency domain, the features were the Power Spectral Density (PSD), Mean Frequency (MNF) and Median Frequency (MDF). The formulas for all the features are presented in Table 1.

#### 2.4.4. Classification Models

To find the best classifier model for this study that performs well in classifying all classes, four commonly used ML methods were selected and applied: K-Nearest Neighbor (KNN) [29], Support Vector Machine (SVM) [30], Random Forest (RF) [31] and Gradient Boosting Machine (GBM) [32]. These four methods were all applied with the goal of classifying the groups into spastic, hypotonic, healthy and rigid. To ensure that the comparison between classifiers was unbiased, parameter optimization was performed for all of them using R-Studio with the R language [28].

Each of the thirty-nine participants performed 15 active movements. After segmenting the sEMG signal into windows, the selected features were extracted, resulting in a total of 585 samples. This study recorded sEMG signals from two muscles (biceps and triceps brachii), so the dataset created is composed of 22 columns for the features (11 for each muscle) and 1 column for the label identifying the type of muscle tone. After creating the datasets, they were divided into a training set and a test set at a ratio of 80:20, and the 10-fold cross-validation method was also performed. The metrics used to evaluate the performance of the classifiers were accuracy (the proportion of correct predictions), precision (the percentage of true positives among all predicted positives), recall (also known as sensitivity, which is the proportion of true positives among all actual positives) and the F1-score (the harmonic mean of precision and recall, balancing both metrics).

## 3. Results

### 3.1. Participants

In this study, 39 participants were included, consisting of 10 with spasticity, 10 with rigidity, 9 with hypotonia and 10 healthy individuals. Table 2 presents the information for all the participants. The age range for inclusion in this study was 18 to 80 years, and the sample comprised participants aged between 28 and 70 years. Although it was not a criterion, all the participants included in the spastic group had a history of ischemic or hemorrhagic stroke affecting either the right or left cerebral hemispheres.

### 3.2. Classification Performance

With the objective of finding the model that best classifies and characterizes the sEMG signal of individuals with abnormalities in muscle tone, we calculated the performance in terms of the accuracy, precision, recall and F1-score of the four models selected for analysis (KNN, RF, GBM and SVM). The optimal parameters of the four classifier models applied in this study are shown in the Appendix A. The results can be seen in Table 3.

We found F1-scores of 0.9628, 0.9136, 0.9249 and 0.9413 for KNN, RF, GBM and SVM, respectively, with an average of 0.9357. This shows that the applied classifiers were highly effective, presenting a good balance between precision and recall, indicating a robust capability to correctly identify the corresponding muscle tone and minimize false positives and false negatives.

Among the classifiers analyzed, the one that presented the best accuracy was KNN, with 96.12% correct classifications, followed by SVM with 94.17%. Figure 3 shows the accuracy of the models for each of the classes, i.e., the hit rate for classifying individuals as healthy, hypotonic, rigid and spastic.

In this analysis, the models that stand out are KNN and SVM, with KNN being the best in all classes. KNN presented an accuracy of 96.4% for healthy, 95.0% for hypotonic, 93.3% for rigid and 100% for spastic individuals. The KNN model successfully classified all spastic individuals but had its worst performance with rigid individuals. The model’s biggest confusion was related to rigid individuals being incorrectly classified as healthy (6.7%). The SVM model achieved 100% accuracy for the healthy group and also showed the lowest accuracy for the rigid individuals class (89.7%). The model with the lowest accuracy of all was RF for rigid individuals (86.2%).

### 3.3. Feature Importance

In addition to evaluating the performance of the analyzed classifiers, we also assessed the importance of the features extracted from the sEMG signal. The feature importance ratio is the sum of the two muscles of each feature, and the results are presented as percentages in Figure 4.

As a result, we found that the most important features for classification were the WL (14.88%) and MAV (14.25%). WL is a feature of the sEMG signal that represents the amount of variation or activity in the signal, being a measure of the complexity of muscle movement. Meanwhile, the MAV provides information about the average amplitude of the signal, reflecting the average intensity of muscle contraction. Thus, it was possible to verify that these signal features had the most impact on the classification among different types of abnormalities of muscle tone (spasticity, normotonicity, hypotonicity and rigidity).

Regarding the importance of muscles in classification accuracy, the results showed that the biceps had an importance of 55.94% and the triceps brachii 44.06%, indicating that the biceps plays a slightly more significant role in the classification of muscle conditions compared to the triceps brachii.

## 4. Discussion

Abnormalities in muscle tone caused by neurological disorders can lead to painful contractures, reduced mobility and, consequently, diminished autonomy for those affected, potentially resulting in a lower quality of life [33]. This study decided to focus on the upper limbs, as they play a fundamental role in performing daily tasks, interacting with the environment and participating in social activities [5]. Specifically, we concentrated on the muscles involved in elbow movement (biceps and triceps brachii), as they are crucial for achieving a wide range of functional activities.

The evaluation and diagnosis of muscle tone alterations are currently performed mainly using ordinal scales, such as the MAS for spasticity and the UPDRS for rigidity. These scales depend on the experience of the therapist applying them, which can lead to discrepancies between evaluations and consequently reduce the reliability of the scale [34]. This underscores the need for methods that can reliably and objectively assess and distinguish these abnormalities of muscle tone.

In this paper, we introduce a new methodology that involves the use of sEMG and ML to classify individuals across the spectrum of muscle tone (spastic, healthy, hypotonic and rigid individuals) in the upper limbs, through active movements, with the aim of addressing this gap in the literature by developing methods that can be applied to various types of muscle tone. The proposed method showed good accuracy results for all the evaluated models and was able to distinguish individuals between the groups.

The classifiers applied in this study were chosen based on previous research that utilized these models for sEMG classification, such as [29], who used KNN, and [32], who used GBM. Most articles in the literature that apply these algorithms for sEMG classification focus on gesture recognition. For comparison, the accuracies found by [30,32] were close to or lower to those found in our study (91% and 83.9%, respectively), showing good compatibility between the results.

The model from our study that achieved the highest accuracy was KNN, with a 96.12% success rate. This algorithm is one of the most widely used in solving various practical problems. KNN has a simple functioning system, as the methodology is based on the idea that if most of the k-nearest samples to a given sample belong to a certain category, the sample also belongs to this category [35]. This simplicity can be advantageous for future applications in real-time systems, as KNN is easier to implement than other evaluated algorithms and performs well when properly optimized.

Despite the high accuracy observed, the analyzed algorithms showed some classification errors. For KNN and SVM, the classifiers with the highest accuracy, the greatest confusion occurred with the “rigid” class, where almost 7% of the data were incorrectly classified as “healthy”. This may have occurred because the patients were evaluated in their normal state, meaning no subject was asked to suspend their medications for the data collection. As a result, the medications of the rigid group may have influenced the degree of rigidity. However, the data collection was conducted with the aim of approximating real clinical practice, where evaluations are performed with patients under the effect of medications.

Chen et al. [20] also used sEMG and ML applied to individuals with muscle tone abnormalities. Their study evaluated the degree of spasticity of the upper limbs in patients based on the MAS. The study achieved classifier accuracies of 98.2%, 90.6%, 95.8% and 92.9% with the RF method for each of the MAS levels (0, 1, 1+ and 2), respectively. Thus, the accuracies found for these classes were close to those found in our study, confirming the good performance of sEMG classification in spastic individuals.

Murias et al. [36] presented a method to evaluate rigidity in PD individuals, using inertial sensors on the wrist. The study achieved an accuracy of 80% in identifying changes in rigidity. On the other hand, ref. [37] used sEMG, but it was not focused on identifying rigidity; instead, it aimed to characterize PD as a whole and achieved an average accuracy of 76.6%. These results show that our study achieved higher accuracy compared to similar studies.

Regarding the most important features for the good accuracy found, the WL and MAV stood out. These features pertain to the complexity of the signal and the average intensity of muscle activity, respectively. The work by [20] also verified the importance of these features, and in the study, WL and MAV were among the top five. This shows that these characteristics can be determining factors in differentiating between types of muscle tone and can be used in future applications and studies.

In our proposed protocol, the objective was to evaluate the upper limb by capturing the sEMG signal from the biceps and triceps brachii. Assessing the importance of each in the classification accuracy between groups, the biceps had 55.94% and the triceps 44.06%. Therefore, we found that in performing active movements, the biceps plays a slightly more significant role in classifying muscle tone compared to the triceps. However, both muscles are necessary for a comprehensive assessment, especially for conditions such as rigidity that affect the entire body [38].

All the results found in this study demonstrate the feasibility of using sEMG and ML to classify and characterize muscle tone abnormalities. These findings, particularly regarding the best algorithm and the most important features, can be embedded in future clinical devices. This would allow for more objective assessments and the measurement of treatment or medication effectiveness, significantly benefiting patients with these conditions.

## 5. Limitations and Suggestions for Future Studies

Despite the promising results, this study has some limitations. First, the sample size was relatively small, which may limit the generalizability of the findings. Another point is the inclusion in the protocol of only active movements, thus restricting the inclusion to individuals capable of performing the movements and excluding those who are plegic.

Future work should aim to expand the number of participants to include a larger and more diverse population, spanning multiple neurological conditions and muscle groups, as well as to enhance the protocol to include passive movements. Longitudinal studies could also be conducted to assess the consistency of the classification models over time, across different clinical conditions, and for monitoring various treatment approaches. Additionally, exploring advanced machine learning techniques, such as deep learning, may improve the model’s ability to capture complex patterns in sEMG signals. Finally, developing user-friendly software tools for clinicians to easily apply these methods in clinical practice would be a valuable step toward translating this research into real-world applications.

## 6. Conclusions

The findings of this study demonstrate the efficacy of sEMG signals for the objective classification and characterization of abnormalities of muscle tone. The implementation of ML algorithms, including KNN, SVM, RF and GBM, yielded high classification accuracies, with KNN achieving the best overall performance. This underscores the potential of sEMG and advanced computational techniques to overcome the limitations of traditional subjective assessments, such as the MAS and UPDRS, by providing a more reliable and quantitative approach for evaluating muscle tone. This study’s results suggest that features like WL and MAV are particularly influential in distinguishing between different types of muscle tone, highlighting the importance of these parameters in future research and clinical applications.

## Figures and Tables

**Figure 1 sensors-24-06362-f001:**
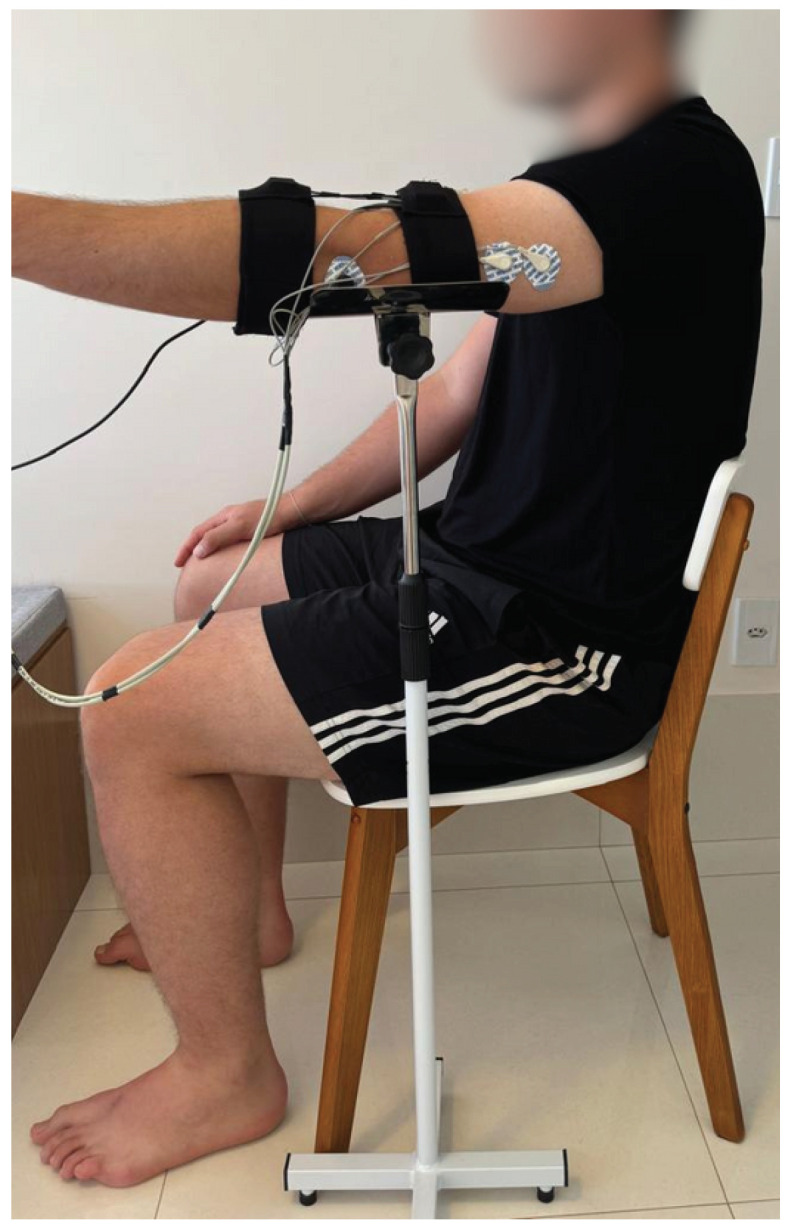
Participant’s arm positioning for data collection.

**Figure 2 sensors-24-06362-f002:**
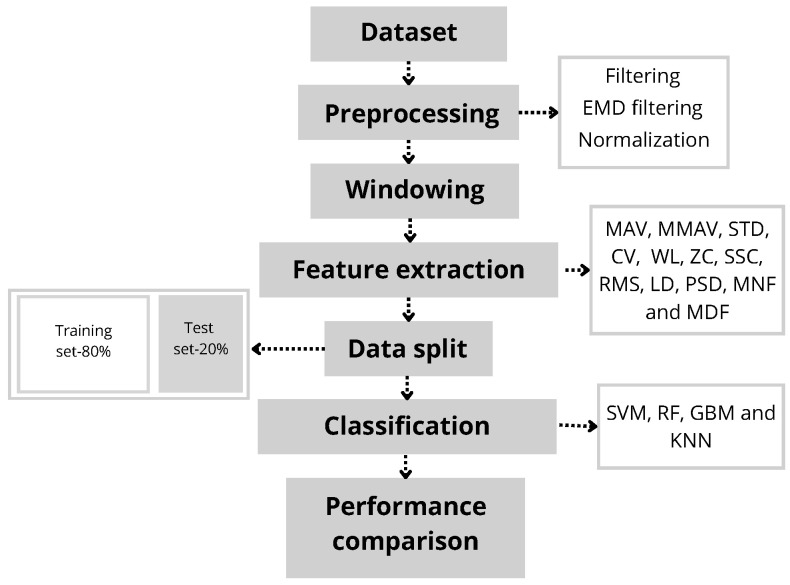
Flowchart of the proposed signal processing. Firstly, the sEMG signals are preprocessed. Then, the windowing and feature extraction are performed on the preprocessed data. Finally, the performance of the four classification models (RF, KNN, SVM and GBM) is verified using 10-fold cross-validation.

**Figure 3 sensors-24-06362-f003:**
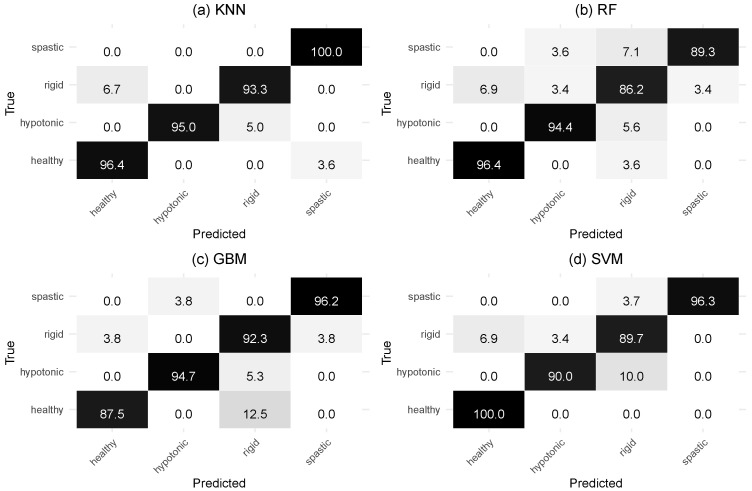
Confusion matrix of classifier models. (**a**) KNN model, (**b**) RF model, (**c**) GBM model and (**d**) SVM model. All the elements of the matrix are presented as percentages. The sum of the elements of each row is equal to 100, and the diagonal of the matrix represents the correct classifications. Darker colors indicate higher values, while lighter colors represent lower values.

**Figure 4 sensors-24-06362-f004:**
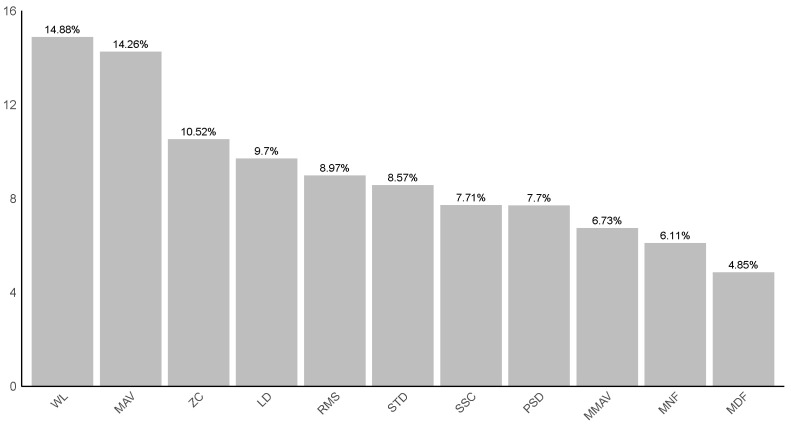
Feature importance. The Y-axis coordinate is the importance ratio, given as a percentage. The sum of the values is 100%.

**Table 1 sensors-24-06362-t001:** Features extracted in the study.

Feature	Formula
Mean Absolute Value (MAV)	$\frac{1}{N}\sum_{i=1}^{N}\left|x_{i}\right|$
Modified Mean Absolute Value (MMAV)	$\frac{1}{M}\sum_{j=1}^{M}\left(\frac{1}{N_{j}}\sum_{i=1}^{N_{j}}\left|x_{i,j}\right|\right)$
Standard Deviation (STD)	$\sqrt{\frac{1}{N}\sum_{i=1}^{N}{\left(x_{i}-\bar{x}\right)}^{2}}$
Waveform Length (WL)	$\sum_{i=1}^{N-1}\left|x_{i+1}-x_{i}\right|$
Zero Crossing (ZL)	$\sum_{i=1}^{N-1}\left[\mathrm{sgn}\left(x_{i}\right)\neq \mathrm{sgn}\left(x_{i+1}\right)\right]$
Slope Sign Change (SSC)	$\frac{1}{N}\sum_{i=1}^{N-2}1\left\{\left(x_{i+1}-x_{i}\right)\cdot \left(x_{i+2}-x_{i+1}\right)<0\right\}$
Root Mean Square (RMS)	$\sqrt{\frac{1}{N}\sum_{i=1}^{N}x_{i}^{2}}$
Log Detector (LD)	$\log\left(\frac{1}{N}\sum_{i=1}^{N}x_{i}^{2}+1\right)$
Mean Frequency (MNF)	$\frac{\sum_{f=1}^{F}f\cdot P\left(f\right)}{\sum_{f=1}^{F}P\left(f\right)}$
Median Frequency (MDF)	The frequency at which the power spectrum is divided into two equal halves
Power Spectral Density (PSD)	$\frac{1}{N}{\left|\sum_{n=0}^{N-1}x_{n}e^{-j2\pi \frac{fn}{N}}\right|}^{2}$

**Table 2 sensors-24-06362-t002:** Main information of the participants included in this study.

Subject	Age	Gender	Diagnosis
Sp01	46	Male	Post-stroke
Sp02	40	Female	Post-stroke
Sp03	67	Male	Post-stroke
Sp04	67	Male	Post-stroke
Sp05	57	Male	Post-stroke
Sp06	54	Male	Post-stroke
Sp07	70	Male	Post-stroke
Sp08	28	Female	Post-stroke
Sp09	63	Male	Post-stroke
Sp10	69	Female	Post-stroke
Hp01	61	Male	Dystrophy
Hp02	69	Female	Post-stroke
Hp03	57	Female	Post-stroke
Hp04	60	Female	Post-stroke
Hp05	53	Male	Dystrophy
Hp06	68	Male	MS
Hp07	67	Male	Post-stroke
Hp08	60	Female	Post-stroke
Hp09	62	Male	Post-stroke
Rg01	63	Male	PD
Rg02	57	Female	PD
Rg03	50	Male	PD
Rg04	62	Female	PD
Rg05	64	Female	PD
Rg06	65	Female	PD
Rg07	63	Female	PD
Rg08	62	Female	PD
Rg09	64	Male	PD
Rg10	67	Male	PD
He01	68	Male	-
He02	57	Male	-
He03	64	Male	-
He04	61	Male	-
He05	65	Male	-
He06	57	Male	-
He07	56	Female	-
He08	58	Female	-
He09	52	Female	-
He10	67	Female	-

SP: spastic group; Hp: hypotonic group; Rg: rigidity group; He: healthy group; PD: Parkinson’s Disease; MS: multiple sclerosis.

**Table 3 sensors-24-06362-t003:** Average performance of classifier models.

Model	Accuracy	Precision	Recall	F1-Score
KNN	0.9612	0.9645	0.9619	0.9628
RF	0.9126	0.9123	0.9159	0.9136
GBM	0.9223	0.9255	0.9267	0.9249
SVM	0.9417	0.9437	0.9399	0.9413

## Data Availability

Data are contained within the article.

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
