# Peer review of "Muscle Tone Assessment by Machine Learning Using Surface Electromyography"

_sensors, 2024, doi:10.3390/s24196362_

Round 1

Reviewer 1 Report

Comments and Suggestions for Authors

This study aims to use surface electromyography and machine learning to provide a more objective and reliable assessment of muscle tone. This objective is clear and aligns with the need for more accurate assessment methods.

The following are recommendations for improvement:

Introduction

Page 2. line 47: Several studies have developed methods to objectively assess different types of muscle tone. These methods typically utilize.....you much check and include also other methods typically utilize:   Tensiomyography, Myotonometry, Ultrasound Elastography or Biomechanical Modeling sensors such as accelerometers or gyroscopes to simulate and measure the dynamic properties of muscle tone during movement or at rest.

You must also introduce Machine Learning.

Methods:

The use of sEMG and machine learning is innovative and suitable for achieving the study’s goals. sEMG can provide detailed data on muscle activity, and machine learning can analyze this data to identify patterns and make predictions.

Table 1 refers to the results section. 

Page 3, line 98: What is the inter-electrode distance?

Results:

The use of machine learning for data analysis is appropriate given the complexity of the data. The study should specify which machine learning algorithms will be used and how their performance will be evaluated.

Comments on the Quality of English Language

English quality of the paper:

  • The text is generally clear, but some sentences could be simplified for better readability.
  • There are minor grammatical errors that need correction.

Reviewer 2 Report

Comments and Suggestions for Authors

1.The Manuscript describes the patient’s age ranging from 18 to 80, but in Table 1, the participant age are shown as 28 to 70.Please revise accordingly.

2.Why were only the biceps and triceps muscles used in the study to assess muscle tone? Whether other muscles were considered, please give reasons.

3.Please add an explanation of the relationship between the four physiological states of spasticity, normal, hypotonia and rigidity.

4.In Section 2.4.2, the EMG is decomposed into multiple windows, may I ask the authors how long a window was used to segment the EMG signal and what sample size was obtained for each experiment.

5.The authors were asked to list in detail the classification results (including accuracy, precision, recall and F1 score terms) of the subjects under 10-fold cross-validation and to calculate the mean and standard deviation for comparative analyses.

6.The Manuscript mentions using 11 features to train and test the classification model. I think it too much, potentially reducing the performance of the classification model. If the model performance is affected, it is recommended to perform feature selection.

6. Both Table 4 and Figure 3 present confusion matrices showing the classification accuracy for each label. It is suggested to keep only one of these.

Comments on the Quality of English Language

good, Minor editing of English language required.
